# Shoulder-Restricted Friction Deposition for Aluminum Alloy Coatings on Titanium Alloys

Yuanhang Guo [1], Jihong Dong [1,*], Zongliang Lv [2], Yuming Xie [2,3], Yongxian Huang [2,3] and Xiangchen Meng [2,3,*]

[1] AVIC Manufacturing Technology Institute, Beijing 100024, China; yhguo1992@163.com
[2] State Key Laboratory of Advanced Welding and Joining, Harbin Institute of Technology, Harbin 150001, China; kkyunhui@163.com (Z.L.); ymxie@hit.edu.cn (Y.X.); yxhuang@hit.edu.cn (Y.H.)
[3] Zhengzhou Research Institute, Harbin Institute of Technology, Zhengzhou 450046, China
* Correspondence: dongjihong2021@163.com (J.D.); mengxch@hit.edu.cn (X.M.)

**Abstract:** In order to solve the problem of a thin deposition layer on the titanium alloy in the traditional friction surfacing process of dissimilar Ti/Al metals, new shoulder-restricted friction deposition (SRFD) equipment was successfully developed by introducing a restricted shoulder. Using a laser to roughen the titanium substrate, the process verification of Al deposition onto TC4 was realized. The material utilization was close to 100%, and a deposition layer with a thickness of 0.8 mm and a strong bonded interface was obtained. The peel strength of the triple-layer deposited joints was 121 MPa.

**Keywords:** coatings; friction deposition; aluminum alloys; titanium alloys; dissimilar interface

## 1. Introduction

Aluminum/titanium joints are increasingly attractive for practical applications in the aerospace and automotive industries, as well as in semiconductor titanium sputtering targets [1–4]. Due to the mismatch in their physical properties and metallurgical properties, the joining of Ti/Al alloys remains a tough task [5–7].

Preparing an aluminum transition layer to convert dissimilar Al/Ti metal connections into Al/Al connections is an effective strategy. A representative strategy for preparing such a transition layer is friction surfacing. During the friction surfacing process, the rotating Al rod undergoes severe plastic deformation with the base material under the action of the axial load. The end of the rod is softened, reaching a plastic state, and is then connected to the Ti substrate [8]. The solid-state connection process has a smaller heat input, effectively suppressing the generation of excessively thick intermetallic compounds, which greatly improves the connection performance [9–12]. An effective connection in the friction surfacing process relies on a sufficiently large axial force. However, excessive axial force causes a mushroom-like flash on the Al bar, which leads to a thinner transition layer and low material utilization [13,14]. The balance of the axial force is difficult with this technology, which increases the complexity of parameter control and testing, which hinders its further application. Moreover, the solid connection between the Al and Ti relies on intermetallic compounds (IMCs) with suitable thicknesses [5,7,15,16]. Without the formation of IMCs, the strength of the joint would be low due to insufficient metallurgical properties [15,17–19], while IMCs that are too thick would make the joint brittle instead [20,21]. Wu et al. investigated if IMCs with suitable thicknesses could be achieved by tailoring the heat input [5]. Taking into account both joint forming and IMC control put forward more stringent requirements for the preparation process.

Here, a new tool with a constrained external shoulder was proposed, which increased the heat generation by the shoulder to promote material plasticization. The bottom of the shoulder was designed with a certain concave taper to provide space to accommodate

plastic materials. Moreover, by treating the surface texture, the metallurgical connection of the joint was changed into a coupled connection of metallurgy and mechanical interlocking. Based on the design above, the deposition of thicker and well-connected layers of Al/Ti can be achieved.

## 2. Materials and Methods

In order to solve the problems of poor bonding of the solid phase interface and a thin surfacing layer in dissimilar Ti/Al metal friction surfacing processes, a new shoulder-restricted friction deposition (SRFD) process was proposed. Figure 1 presents a schematic illustration of the SRFD process. The rotating Al rod was first rubbed against the Ti substrate and gradually plasticized. The softened material, under the affection of continued heat and pressure, filled the inner cone cavity at the end of the shoulder. When the inner cavity was filled fully, in addition to the heat generated by the deposited rod and the substrate, friction heat was generated between the plasticized materials themselves and also between the shoulder and the plasticized materials. As the tool traveled, the softened material was stably deposited on the substrate under the action of axial pressure.

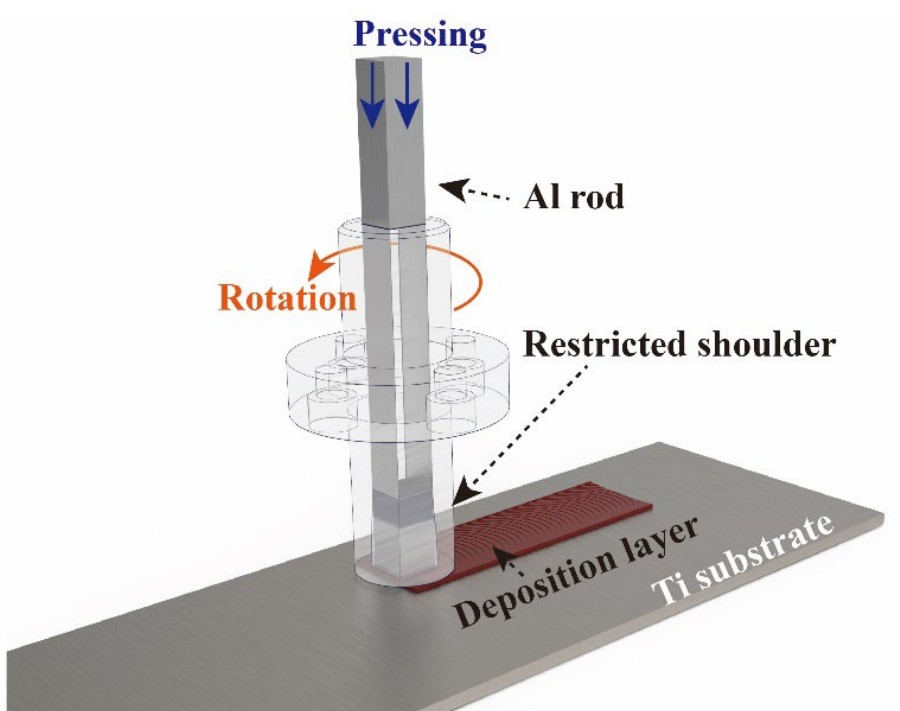

**Figure 1.** Schematic diagram of SRFD process.

The experiment was conducted using a 1064 nm Dapeng laser marking machine (FB-60-GS, Guangzhou, China) tuned to a power output of 60 W, a spot size of 50 μm, and a pulse width of 30 ns. A mixed acid of HF (12 g/L) and $HNO_3$ (175 g/L) was used to remove potential oxides after the surface texturing process. A cross-section of the deposited layer was characterized with an optical microscope (OM, Keyence VHX-7000, Japan). The corroded morphologies of the deposited layer were characterized using a scanning electron microscope (SEM, ZEISS Supra 55, Germany) equipped with an energy dispersive spectrometer (EDS). The grain sizes were measured by electron backscatter diffraction (EBSD) with a step size of 0.5 μm. The specimen for EBSD was polished by a cross-section polisher for 5 h. HKL Channel 5 software was used to analyze the raw EBSD data. A Peel strength test was conducted on an tension tester (AGSX500N-type, Japan) with a punch peeling rate of 0.5 mm/min. A schematic diagram of the punch peeling test is shown in Figure 2, where deposited aluminum with a 10 mm diameter blind hole was

applied to the thrust force by the thrust rod. The interface binding strength was expressed by the ratio of the maximum applied force to the interface binding area.

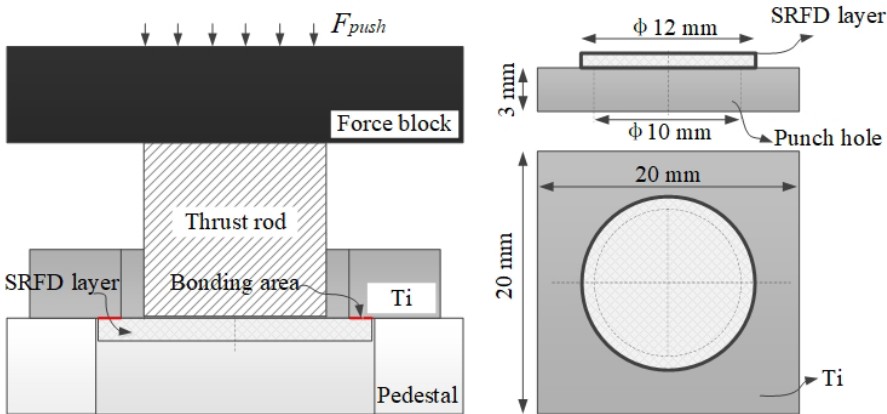

**Figure 2.** Punch peel test process.

## 3. Results and Discussion

### 3.1. Feasibility Analysis of the Designed Tool

In order to promote the deposition of Al on the Ti surface, a laser marking machine was used to perform surface texturing on the Ti surface with a frequency of 50 kHz. Crater-like micro-pits were achieved, as shown in Figure 3.

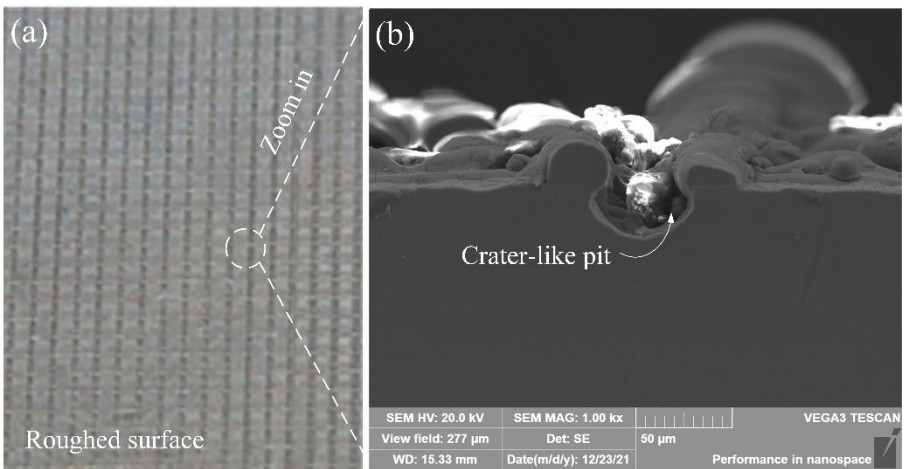

**Figure 3.** Morphology of the surface of titanium alloy after texturing. (**a**) Macro-morphology of the textured surface; (**b**) micro-morphology of the textured surface.

The designed outer diameter of the tool shoulder was 25 mm, and the side length of the central square inner hole was 10 mm. In order to reduce the material feeding resistance, a certain internal cone angle was set at the bottom of the tool, which also provided storage space for the plasticized material.

Assuming that the deposition efficiency was 100%, the material feeding amount was the same as the deposition amount. This was

$$d * h * V_t = S * V_f \tag{1}$$

where $d$ is the diameter of the outer shoulder; $h$ is the lift height of the tool per deposition layer; $V_t$ is the traverse speed of the tool; $S$ is the cross-sectional area of the Al rod; and $V_f$ is the materials' feed rates.

The *d* and *S* in this investigation were 24 mm and 100 mm², respectively. When setting the shoulder lift height to *h* = 0.8 mm, the following can be concluded:

$$V_t = 5 * V_f \tag{2}$$

We set the rotation speed to 1800 rpm, the traveling speed to 1.67 mm/s, and the feeding speed to 0.33 mm/s. Due to errors, such as assembly gaps during the friction deposition process, the feeding speed needed to be compensated to 0.4 mm/s.

It can be seen from Figure 4 that no effective deposition was formed within the first 40 mm, but from 40 mm onwards, a stable deposition layer was gradually formed. In the initial stage, the softened material was needed to fill the gap between the tool and the Ti plate and the gaps between the tool and the Al rod. Only when the closed space among the tool, the Al rod, and the Ti plate was formed could stable deposition be achieved. If there was no shoulder, the heat would be generated only by the friction between the Al rod and the Ti plate. The plasticized material would be rolled up or overflowed in the form of a flash without circumferential constraints. The rotating shoulder with a hole of a certain internal cone angle provided space for material storage, and the plasticized material had circumferential constraints. As shown in Figure 4, there were no obvious mushroom-like flashes around the deposition tool, which preliminarily proved that the material utilization rate was nearly 100%, achieving the expected goal of the tool design.

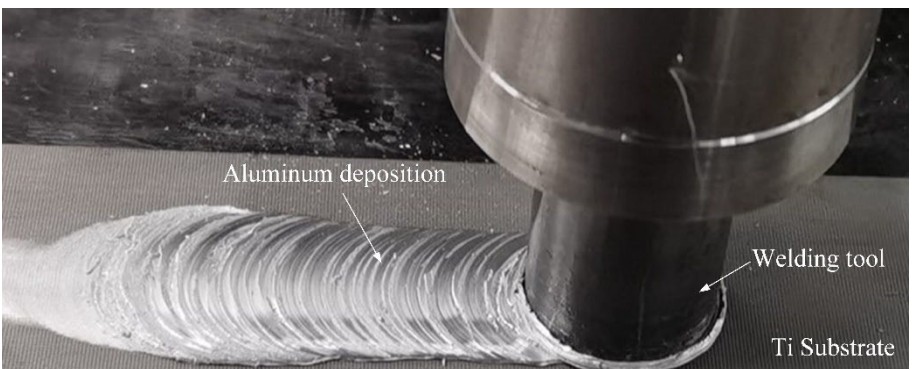

**Figure 4.** Verification of Ti/Al deposition by SRFD equipment.

### 3.2. Microstructural Analysis

Figure 5 presents the cross-sectional morphology of the friction additive joint. According to the characteristics of the deposition process, the joint was divided into a composite deposition zone (CDZ), a shoulder deposition zone (SDZ), and an interfacial zone (IFZ). Great interfacial bonding was generated between the BM and deposition and did not separate during the subsequent deposition, suggesting a metallurgical reaction occurred between the Ti/Al interface.

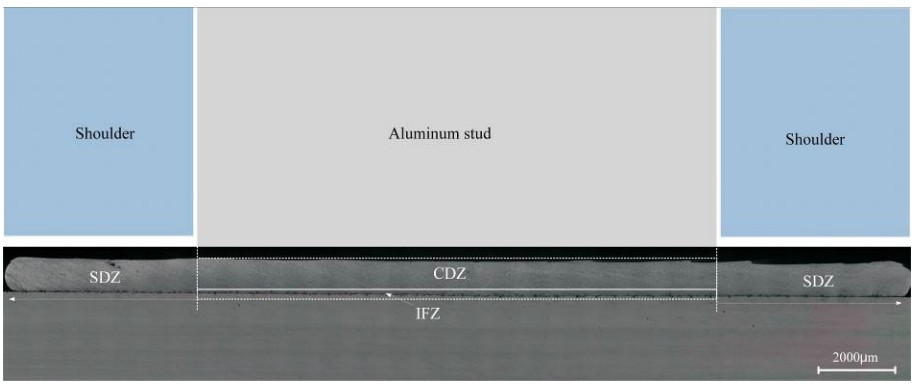

**Figure 5.** The positional relationship of each area of the friction additive joint.

The emergency stop technique of the tool was used to investigate the instantaneous microstructure during the friction deposition process. As shown in Figure 6a, the stationary shoulder restricted the thermo-plasticized material from the flash defect to deposition. Thus, the thickness of the deposited layer was 0.8 mm, consistent with the distance between the stationary shoulder and the substrate. Figure 6c–f present the EBSD results of the CDZ. The sampling position is shown in Figure 6b. Figure 6d is the grain boundary map. Grain boundaries less than 15 were defined as low-angle grain boundaries (LAGBs) and are marked by red lines, and grain boundaries more than 15 were defined as high-angle grain boundaries (HAGBs) and are marked by black lines. It can be seen that the LAGBs were concentrated along the HAGBs. The consequent dislocation generation, pile-up, and annihilation during severe plastic deformation was the symbol of dynamic recrystallization (DRX). Figure 6f displays an IPF map of the CDZ. Around the interface (within 100 μm), the neighboring grain presented slight grain-orientation differences, while the apparent texture emerged in the interlayer and was distributed along the <101> plane. It is well accepted that the CDZ was composed of fine and random DRXed grain. However, the subsequent forging of the stationary shoulder promoted DRXed grain slipping and rotating along a specific crystal face, causing the slip direction to converge to the direction of the principal strain and ultimately leading to a convergence of the grain orientation. The high KAM values in Figure 6e were also attributed to the forging of the stationary shoulder, which could prove the above conclusion.

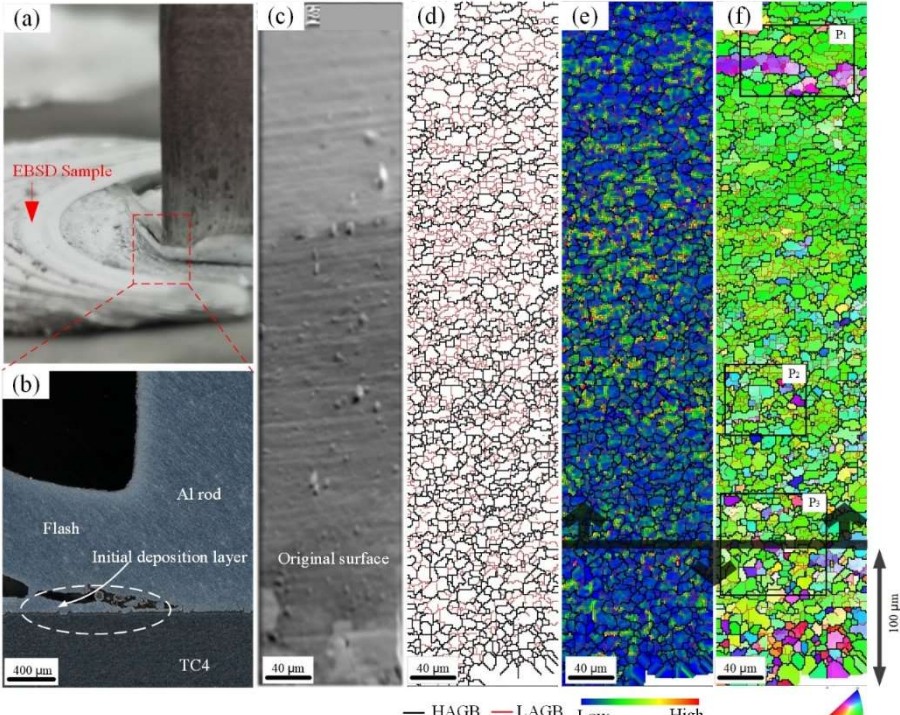

**Figure 6.** EBSD results of the CDZ: (**a**) the sampling position, (**b**) cross-section morphology, (**c**) SEM map, (**d**) grain boundary map, (**e**) KAM map, and (**f**) IPF map.

Figure 7 displays a magnified IPF map and the corresponding grain size distribution of different regions in the CDZ. The average grain size decreased from 11.5 μm in the P1 region to 8.7 μm in the P3 region. This indicates that the dynamic crystallization was more severe near the interface where the friction and plastic deformation occurred between the deposition and the base metal. In addition, laminated fine-grain layers were observed near the interface, consistent with the material flow. The continuous transition during friction deposition led the thin layer to suffer more local plastic deformation and form fine grains with a large number of HAGBs. The stationary shoulder provided forging pressure to enable the formation of the deposited layer and eliminate the flash defect. The

plastic deformation and dynamic crystallization were weaker than those in other regions. Therefore, the grain size of the top region was the largest.

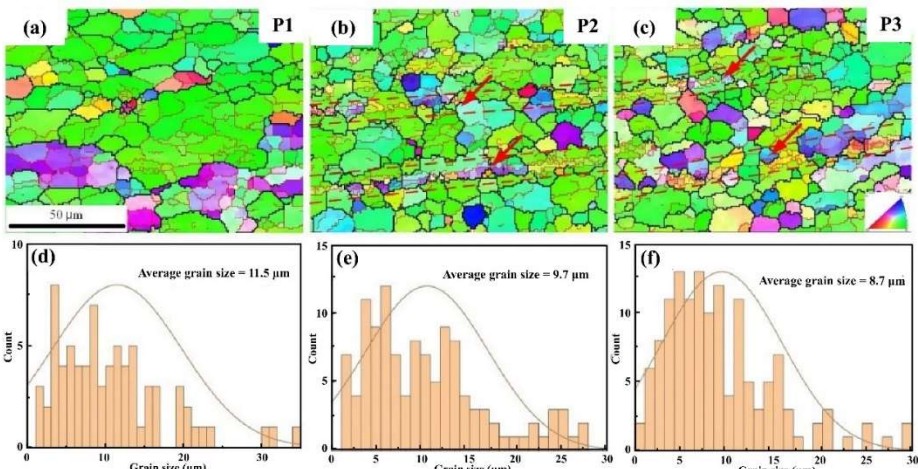

**Figure 7.** Grain size analysis at different positions of the CDZ: (**a**–**c**) P1–P3 in Figure 6f and (**d**–**f**) the corresponding grain size distribution.

Figure 8 presents the morphology of the SDZ. The deposition was completely forged by the stationary shoulder, which recompacted the flash defect. The forging effect also promoted the metallurgical bonding at the Ti/Al interface. Thus, the SDZ morphology was similar to that of the CDZ. However, due to a lack of feedstock stirring, the SDZ presented porosity and kiss-bonding defects. As the thermo-plasticized material accumulated, it formed a semi-confined space with the stationary shoulder and the Ti substrate. The accumulated material built up in this space, producing a passive force exerted by the stationary shoulder. Such force was not sufficient to deposit the thermo-plasticized material. Thus, the SDZ possessed a non-dense microstructure.

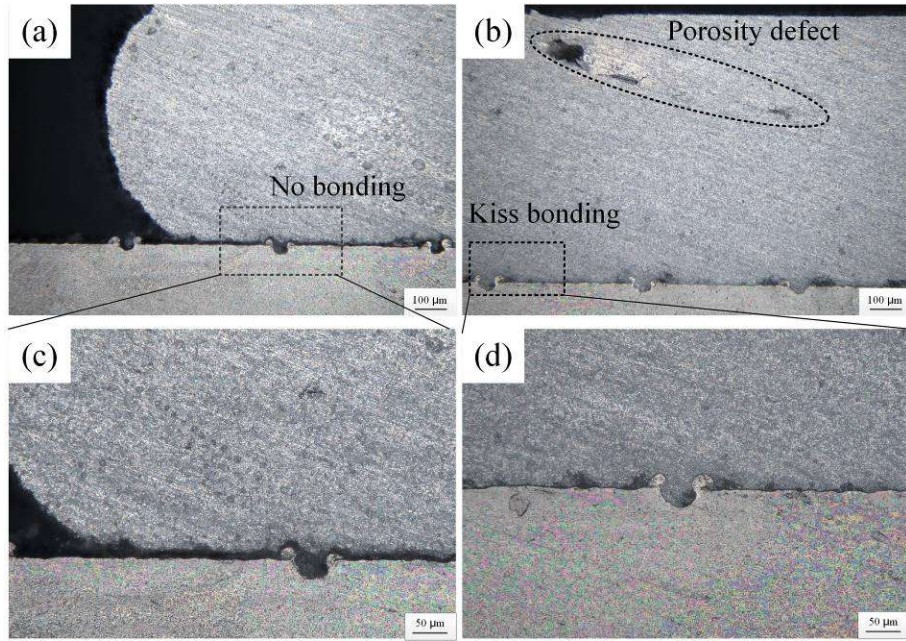

**Figure 8.** Morphology of different positions in the SDZ: (**a**) no bonding edge, (**b**) kiss bonding, (**c**) unfilled in surface texture, and (**d**) magnified map.

The IPZ is the interfacial bonding region formed by the combined reaction between Ti and Al. Due to the high-speed rotation and friction between the feedstock and the substrate,

the substrate in the interfacial zone also underwent severe plastic deformation. Thus, the thickness of the deformed layer was more than 10 μm. The deformation of the Ti in the interfacial zone decreased, and the thickness of the IPZ was only 2 μm in the transition region between the CDZ and the SDZ and presented a discontinuous bonded interfacial layer. It was thought that the heat input away from the center of the feedstock decreased and was not favorable for the interfacial metallurgical reaction. Near the center of the SDZ, the interface showed a tightly connected state (Figure 9c). Near the edge of the SDZ, the axial pressure was released by the open space, and the interface here appeared to be separated (Figure 9d). The EDS scanning result showed a diffusion layer depth of 2 mm in the Ti/Al interface, revealing metallurgical bonding at the interface (Figure 10).

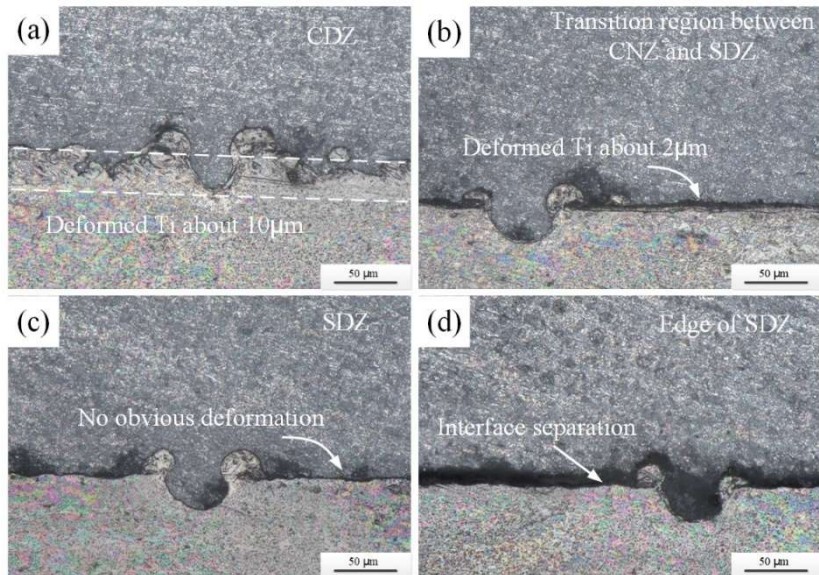

**Figure 9.** Morphology of different positions in the IFZ: (**a**) the interface of the CDZ, (**b**) the transition between the CDZ and the SDZ, (**c**) the SDZ, and (**d**) the edge of the SDZ.

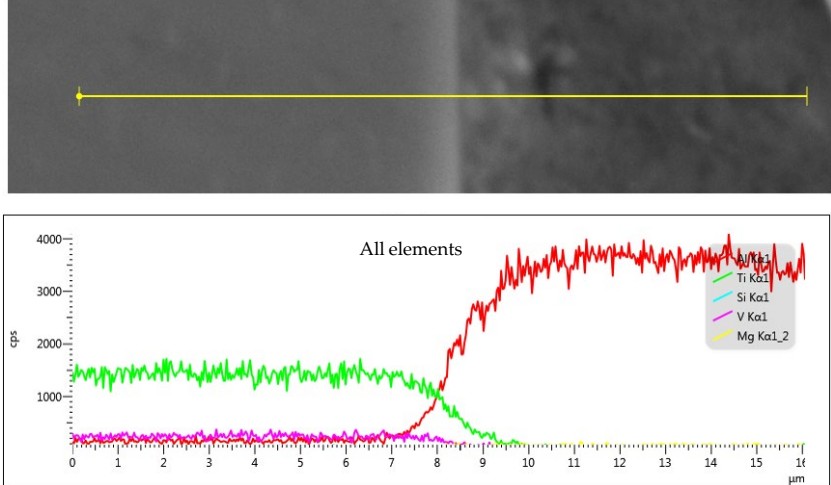

**Figure 10.** EDS scanning at the interface of the CDZ.

### 3.3. Interface Bonding

The thickness of the deposited aluminum on the surface of the matrix was thin, and the bonding strength between the aluminum and the matrix was difficult to measure with compressive shear. The interface bonding can be characterized by a special peel strength test. A square sample with a size of 20 mm × 20 mm was designed, the deposited layer was milled into a circular table with a diameter of 12 mm, and a blind hole with a diameter

of 10 mm was machined from the center of the titanium alloy back until the deposited aluminum was reached. The samples for the punch peeling test are shown in Figure 11, and the punch peel test process is shown in Figure 2. During the test, a round rod with a diameter of 8 mm was applied slowly to the deposited layer until the deposited layer was separated from the titanium alloy.

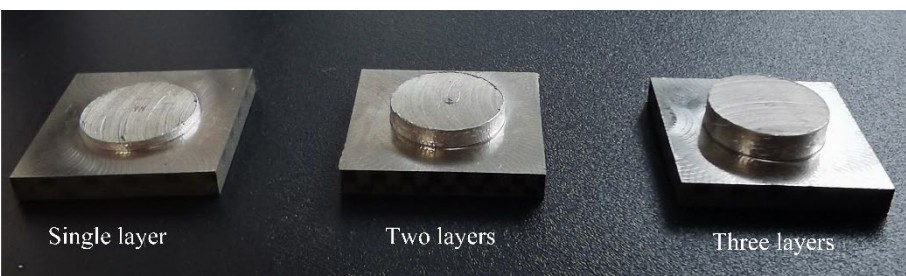

**Figure 11.** Punch peel test samples.

The peel strengths of single-layer, double-layer, and triple-layer deposited joints were 102 MPa, 114 MPa, and 121 MPa, respectively. The increase in the strength was due to the influence of multiple pressure and heat inputs during the subsequent deposition process, which improved the interface bonding strength. Multiple pressure and heat inputs promote sufficient material flow. Thus, the further-softened plasticized material can fill the textured holes well and form solid metallurgical connections under a certain degree of the forging effect.

The peeling process needed to overcome the mechanical interlocking of laser texturing and the metallurgical bonding force between Ti and Al atoms. According to the different binding forces, the fracture appeared to have a different morphology. Figure 12 shows the macro- and micromorphology of the fracture of the double-layer deposited joint. When the interface binding force was high, the peel path extended to the Al side, causing some Al to remain on the surface of the Ti side. Tear edges were observed in the fracture morphology. When the interface bonding force was poor, the peeling surface was the Al/Ti surface. Al alloy plowing behavior may occur in the laser texturing location, resulting in some of the Al alloy remaining in the pits. The fracture morphology of the region without the laser texturing process was relatively flat. The fracture at the cross location of the laser texturing line presented a large, dimpled size. The fracture in the pit presented a typical ductile fracture. The peel strength of the single-layer joint was particularly high. Based on the above analysis, it can be seen that dissimilar Ti/Al welding can obtain a high quality using the SRFD process.

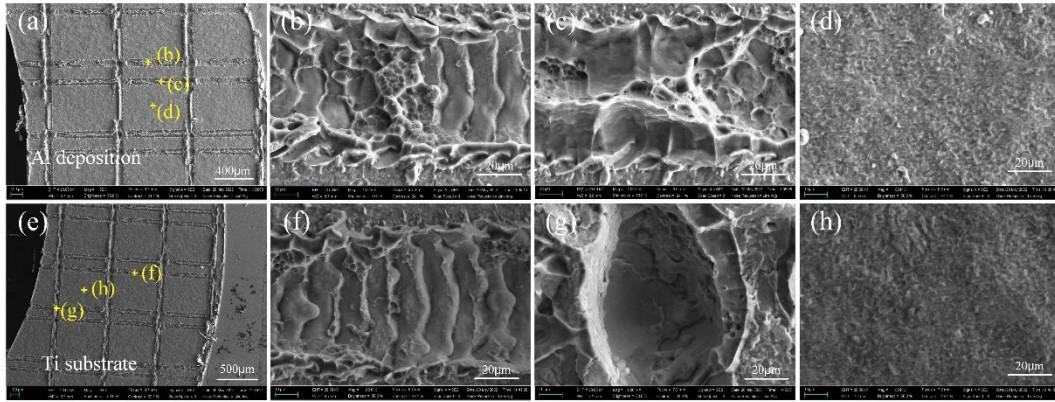

**Figure 12.** Macro/micro-morphology of the fracture surface of double-layer SRFD joint. (**a**) Macromorphology fracture of the Al side; (**b–d**) enlarged fracture characteristics corresponding to b, c, and d in Figure 12a; (**e**) macro-morphology fracture of the Ti side; (**f–h**) enlarged fracture characteristics corresponding to f, g, and h in Figure 12e.

## 4. Conclusions

　　A new SRFD method was proposed to prepare aluminum alloy coatings on the surface of titanium alloys. A well-bonded interface between the aluminum alloy and titanium alloy was formed, while the deposition layer was characterized by fine grains with a size range from 8.7 to 11.5 μm. Under the thermo-mechanical effects, the plasticized materials were squeezed into the micro-grooves induced by laser texturing, forming micro-mechanical interlocking. The peel strength of the triple-layer deposited joints was 121 MPa, which validated that the SRFD process is effective at producing a strong Al/Ti interface.

**Author Contributions:** Conceptualization, X.M. and Z.L.; methodology, Z.L.; validation, Y.X.; formal analysis, Y.H.; investigation, Y.G.; resources, J.D.; writing—original draft preparation, Y.G.; writing—review and editing, Y.G.; visualization, J.D.; funding acquisition, J.D., Y.H. and X.M. All authors have read and agreed to the published version of the manuscript.

**Funding:** This research was funded by the National Natural Science Foundation of China (Nos. 52175301, 52205350, 52305345) and the Special Fund for Research on National Major Research Instruments (No. 52227807).

**Institutional Review Board Statement:** Not applicable.

**Informed Consent Statement:** Not applicable.

**Data Availability Statement:** Data are contained within the article.

**Conflicts of Interest:** The authors declare no conflict of interest.

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
