# Peer review of "Shoulder-Restricted Friction Deposition for Aluminum Alloy Coatings on Titanium Alloys"

_coatings, doi:10.3390/coatings14010130_

Round 1

Reviewer 1 Report

Comments and Suggestions for Authors

The authors propose a novel SFRD method to address the challenges of the Ti/Al dissimilar metal friction surfacing process. Despite the well-structured manuscript, addressing these points would significantly enhance the paper's clarity, depth, and overall potential acceptance in the 'MDPI Coatings.'

1.     In the introduction, consider providing a more seamless connection between the existing problems and how the proposed tool will overcome them.

2.     Explain how the constrained external shoulder increases heat generation and promotes material plasticization.

3.     Show the arrow clearly to label restricted and deposition shoulder in Fig 1.

4.     In section 2, The description of the SRFD process is very generic. Provide a step-by-step breakdown of the process, mainly focusing on the role of the constrained shoulder in enhancing heat input, promoting material flow, and improving interface bonding.

5.     Include a brief justification or rationale for choosing specific testing conditions, such as the 0.5 mm/min punch peeling rate.

6.     Consider rephrasing this sentence for clarity: "The friction heat between the consumable rod and the substrate caused the plasticized material to produce a certain thickness of the deposited layer on the substrate." Make it relatively concise yet explicit.

7.     In section 3.1, it is essential to discuss further the significance of the observed absence of mushroom-like flashes in terms of material utilization, and achieving the expected goal of the tool design is pretty weak.

8.     Regarding microstructural analysis, connect the observed microstructural features to the underlying deposition process. For example, explain how the forging effect of the stationary shoulder contributes to the observed microstructure.

9.     Clarify the specific aspects of the particular peel strength test. Please explain how the test characterizes interface bonding and its suitability for the thin aluminum layer.

10. In section 3.3, the increase in peel strength with the number of layers is discussed very well. However, the authors should explain why multiple pressure and heat inputs enhance interface bonding.

11. What do the authors imply from the different fracture morphologies for the interface bonding quality?

12. Consider adding a sentence or two suggesting potential avenues for future research or improvements to the SRFD method.

13. Additionally, I recommend thoroughly reviewing the article's language to ensure it is free from grammatical errors. While the text is generally clear, I observed a few instances where grammatical corrections might be needed.

Comments on the Quality of English Language

I recommend thoroughly reviewing the article's language to ensure it is free from grammatical errors. While the text is generally clear, I observed a few instances where grammatical corrections might be needed.

Author Response

Dear Editor/Reviewers,

Thank you for the valuable comments on the manuscript entitled "Shoulder restricted friction deposition towards aluminum alloy coatings on the titanium alloy." We really appreciate your carefulness and conscientiousness. We hope that the updated manuscript will meet the high requirements of this journal. You will find our point-by-point responses to the editor and reviewer’s comments below. The modified manuscript has been attached, and all the modifications are highlighted in red. Thank you very much.

Reviewer 1#:

The authors propose a novel SFRD method to address the challenges of the Ti/Al dissimilar metal friction surfacing process. Despite the well-structured manuscript, addressing these points would significantly enhance the paper's clarity, depth, and overall potential acceptance in the 'MDPI Coatings.'

Comments 1: In the introduction, consider providing a more seamless connection between the existing problems and how the proposed tool will overcome them.

Response to comment 1

Thanks for the comments, which were very helpful in improving our manuscript.

The Introduction section has been rewritten, and the following content has been added (highlighted in the manuscript).

“Moreover, the solid connection between the Al/Ti has relied on the intermetallic compounds (IMCs) with suitable thickness[5,7,15,16]. Without the formation of IMCs, the strength of the joint would be low due to insufficient metallurgical[15,17–19]. While too thick IMC would make the joint brittle instead[20,21]. Wu et al. investigated that IMCs with suitable thickness could be achieved by tailoring the heat input[5]. Taking into account both joint forming and IMC control put forward more stringent requirements for the preparation process.

Here, a new tool with a constrained external shoulder was proposed, which increased the heat generation by the shoulder to promote material plasticization. The bottom of the shoulder was designed with a certain concave taper to provide space for accommodating plastic materials. Moreover, with surface texturing treated, the metallurgical connection of the joint was changed into a coupled connection of metallurgy and mechanical interlocking. Based on the design above, deposition of thicker and well-connected layers of Al/Ti can be achieved.

Comments 2: Explain how the constrained external shoulder increases heat generation and promotes material plasticization.

Response to comment 2

Thanks for the comments. If there were no shoulders, the heat would be generated only by the friction between the rod and the Ti base plate. The plasticized material was rolled up or overflowed in the form of a flash without circumferential constraints, while the rotating shoulder with a hole of a certain internal cone angle provided space for material storage, and the plasticized material had circumferential constraints. Thus, in addition to the heat generated by the deposited rod and the substrate, friction heat was generated between the plasticized materials themselves and also between the shoulder and the plasticized materials. Therefore, more heat was obtained, which was more conducive to the plasticization of materials.

The related discussion was added to the manuscript.

Comments 3: Show the arrow clearly to label restricted and deposition shoulder in Fig1.

Response to comment 3

Thanks for the comments. The Fig. 1 was rearranged with a clear label, as shown below.

Comments 4: In section 2, The description of the SRFD process is very generic. Provide a step-by-step breakdown of the process, mainly focusing on the role of the constrained shoulder in enhancing heat input, promoting material flow, and improving interface bonding.

Response to comment 4

Thanks for the comments. The description of the SRFD process was discussed in detail, as follows:

“The rotating Al rod was first rubbed against the Ti substrate and gradually plasticized. The softened material, under the action of continued heat and pressure, filled the inner cone cavity at the end of the shoulder. When the inner cavity was filled fully, in addition to the heat generated by the deposited rod and the substrate, friction heat was also generated between the plasticized materials themselves and between the shoulder and the plasticized materials. As the tool traveled, the softened material was stably deposited on the substrate under the action of axial pressure.”

Comments 5: Include a brief justification or rationale for choosing specific testing conditions, such as the 0.5 mm/min punch peeling rate.

Response to comment 5

Thanks for the comments. The deposition layer on the surface of the matrix was thin. Especially the thickness of the single layer was less than 1 mm, and the interface bonding strength can not be characterized by the conventional compression shear method. Detailed information was added to the manuscript.

Comments 6: Consider rephrasing this sentence for clarity: "The friction heat between the consumable rod and the substrate caused the plasticized material to produce a certain thickness of the deposited layer on the substrate." Make it relatively concise yet explicit.

Response to comment 6

Thanks for the comments. As the reviewer mentioned in point 4, the related part was rewritten. And the “The friction…….. layer on the substrate” was deleted and rewritten also. The added part was shown in response in point 4 and was highlighted in the manuscript.

Comments 7: In section 3.1, it is essential to discuss further the significance of the observed absence of mushroom-like flashes in terms of material utilization, and achieving the expected goal of the tool design is pretty weak.

Response to comment 7

Thanks for the comments. The related part was further discussed. The revised part was highlighted in the manuscript as following: “If there were no shoulder, the heat would be generated only by the friction between the Al rod and the Ti plate. The plasticized material would be rolled up or overflowed in the form of a flash without circumferential constraints, while the rotating shoulder with a hole of a certain internal cone angle provided space for material storage, and the plasticized material had circumferential constraints. As shown in Fig. 3, there were no obvious mushroom-like flashes around the deposition tool, which preliminarily proved that the material utilization rate was nearly 100%, achieving the expected goal of tool design.”

Comments 8:Regarding microstructural analysis, connect the observed microstructural features to the underlying deposition process. For example, explain how the forging effect of the stationary shoulder contributes to the observed microstructure.

Response to comment 8

Thanks for the comments. The average grain size was decreased from 11.5 μm of the P1 region to 8.7 μm of the P3 region. It indicated that the dynamic crystallization was more severe near the interface where the friction and plastic deformation occurred between the deposition and the base metal. The stationary shoulder provided forging pressure to enable the formation of the deposited layer and eliminate the flash defect. The plastic deformation and dynamic crystallization were weaker than other regions. Therefore, the grain size of the top region was the largest. We have added these sentences to the manuscript.

Comments 9: Clarify the specific aspects of the particular peel strength test. Please explain how the test characterizes interface bonding and its suitability for the thin aluminum layer.

Response to comment 9

Thanks for the comments. The deposited aluminum with a 10 mm diameter blind hole was applied to the thrust force by the thrust rod. The interface binding strength was expressed by the ratio of the maximum applied force to the interface binding area. When the thickness of deposited aluminum was less than 2 mm, this method can be used to character the interfacial bonding strength between the deposited layer and the matrix. Detailed information was given in the manuscript.

Fig. 2 Punch peel test process

Comments 10: In section 3.3, the increase in peel strength with the number of layers is discussed very well. However, the authors should explain why multiple pressure and heat inputs enhance interface bonding.

Response to comment 10

Thanks for the comments. Multiple pressure and heat inputs promote sufficient material flow. Thus, the further softened plasticized material could fill the textured holes well and form solid metallurgical connections under a certain degree of the forging effect.

Comments 11: What do the authors imply from the different fracture morphologies for the interface bonding quality?

Response to comment 11

Thanks for the comments. Large and deep dimples can be observed on the aluminum side where the fracture occurs, indicating high bonding quality between the deposited aluminum and the matrix. Shallow and small dimples can be observed at the fracture location of the Ti/Al interface, and the bonding strength of the interface is poor.

Comments 12: Consider adding a sentence or two suggesting potential avenues for future research or improvements to the SRFD method.

Response to comment 12

Thanks for the comments. The deposited material acts as a transition layer via the SRFD method to convert dissimilar metal connections into connections with the same materials.

Comments 13:Additionally, I recommend thoroughly reviewing the article's language to ensure it is free from grammatical errors. While the text is generally clear, I observed a few instances where grammatical corrections might be needed.

I recommend thoroughly reviewing the article's language to ensure it is free from grammatical errors. While the text is generally clear, I observed a few instances where grammatical corrections might be needed.

Response to comment 13

Thanks for the comments. The whole manuscript was modified and checked twice.

Reviewer 2 Report

Comments and Suggestions for Authors

The introduction is treated very superficial. Noticed only 11 references, in a field where continuum development occur as even the authors stated, seems very low which endorse my view about superficiality of literature review

What about the use of this technique proposed here for industrial applications ? please clearly state the benefits and weakness

Details of samples preparation for different measurement indicated in the methods should be carefully escribed for reproduction; also details of data processing and post processing requires to be detailed in the manuscript. For example, how I can reproduce the EBSD samples, measurements and post processing and with which software ? A detailed description is required

What about potential oxidation during laser texture and afterwords ?

A proper details of material transition is required – for now the results especially figure 5 b against figure 5f do not agree – otherwise through black line the authors indicate a line far away of “welding zone”

Figure 6 is of poor quality

Some more appropriate discussion against literature data is required

Conclusion are treated very superficial and more quantitative details are required

The state of art and reference list is limited

Comments on the Quality of English Language

Extensive editing of English language required

Author Response

Dear Editor/Reviewers,

Thank you for the valuable comments on the manuscript entitled "Shoulder restricted friction deposition towards aluminum alloy coatings on the titanium alloy." We really appreciate your carefulness and conscientiousness. We hope that the updated manuscript will meet the high requirements of this journal. You will find our point-by-point responses to the editor and reviewer’s comments below. The modified manuscript has been attached, and all the modifications are highlighted in red. Thank you very much.

Comments 1: The introduction is treated very superficial. Noticed only 11 references, in a field where continuum development occur as even the authors stated, seems very low which endorse my view about superficiality of literature review. What about the use of this technique proposed here for industrial applications ? please clearly state the benefits and weakness

Response to comment 1

Thanks for the comments. The introduction was written and added the related discussion, which was highlighted in the manuscript.

The added part was as follows: “Moreover, the solid connection between the Al/Ti has relied on the intermetallic compounds (IMCs) with suitable thickness[5,7,15,16]. Without the formation of IMCs, the strength of the joint would be low due to insufficient metallurgical[15,17–19]. While too thick IMC would make the joint brittle instead[20,21]. Wu et al. investigated that IMCs with suitable thickness could be achieved by tailoring the heat input[5]. Taking into account both joint forming and IMC control put forward more stringent requirements for the preparation process.

Here, a new tool with a constrained external shoulder was proposed, which increased the heat generation by the shoulder to promote material plasticization. The bottom of the shoulder was designed with a certain concave taper to provide space for accommodating plastic materials. Moreover, with surface texturing treated, the metallurgical connection of the joint was changed into a coupled connection of metallurgy and mechanical interlocking. Based on the design above, deposition of thicker and well-connected layers of Al/Ti can be achieved.”.

Comments 2: Details of sample preparation for different measurement indicated in the methods should be carefully escribed for reproduction; also details of data processing and post processing requires to be detailed in the manuscript. For example, how I can reproduce the EBSD samples, measurements and post processing and with which software ? A detailed description is required

Response to comment 2

Thanks for the comments. This section has been revised, and the detailed content was given in the materials and methods section, such as:

“The experiment was conducted by a 1064 nm Dapeng laser marking machine (FB-60-GS) tuned to a power output of 60 W, a spot size of 50 μm, and a pulse width of 30 ns. The mixed acid of HF(12 g/L) and HNO3(175 g/L) was used to remove potential oxides after the surface texturing process. The specimen for EBSD was polished by a cross-section polisher for 5 h. The HKL Channel 5 software was used to analyze the raw EBSD data. A Peel strength test was conducted on an AGSX500N-type tension tester with a punch peeling rate of 0.5 mm/min. The schematic diagram of the punch peeling test is shown in Fig. 2, where the deposited aluminum with a 10 mm diameter blind hole was applied to the thrust force by the thrust rod. The interface binding strength was expressed by the ratio of the maximum applied force to the interface binding area.”

Fig. 2 Punch peel test process

Comments 3: What about potential oxidation during laser texture and afterward?

Response to comment 3

Thanks for the comments. The surface texturing process was protected by an argon atmosphere. After the mixed acid of HF(12 g/L) and HNO3(175 g/L) was used to remove potential oxides. Then, the textured Ti plate was cleaned with alcohol. Specific processing methods were added in the "materials and methods" section.

Comments 4: Proper details of material transition is required – for now the results especially figure 5 b against figure 5f do not agree – otherwise through black line the authors indicate a line far away of “welding zone”

Response to comment 4

Thank you for the advice. The black line was used to highlight the interface of the deposited aluminum and titanium alloy. P1, P2, and P3 were three regions that gradually approached the interface. The average grain size was decreased from 11.5 μm of the P1 region to 8.7 μm of the P3 region. It indicated that the dynamic crystallization was more severe near the interface where the friction and plastic deformation occurred between the deposition and the base metal. We have added these sentences in Section 3.1.

Comments 5: Figure 6 is of poor quality.

Response to comment 5

Thanks for the comments. Thank you for the advice, we have updated Figure 6.

Comments 6: Some more appropriate discussion against literature data is required. Conclusion are treated very superficial and more quantitative details are required. The state of art and reference list is limited. Extensive editing of English language required.

Response to comment 6

Thanks for the comments. Some more data was added and discussed, which was highlighted in red in the manuscript. The conclusion was modified as follows:

 “A new SRFD method was proposed to prepare the aluminum alloy coatings on the surface of titanium alloys. The well-bonded interface between aluminum alloy and titanium alloy was formed, while the deposition layer was characterized by fine grains with a size range from 8.7-11.5 μm. Under the thermo-mechanical effects, the plasticized materials were squeezed into the micro-grooves induced by laser texturing, forming micro-mechanical interlocking. The peel strength of the triple-layer deposited joints was 121 MPa, which validated that the SRFD is effective in producing a strong Al/Ti interface.”

The reference and the English language were edited and checked twice.

We appreciate the Editor and Reviewers’ warm and positive comments earnestly and hope that the revision will meet with approval. Thanks for your nice work again!

Sincerely yours,

Xiangchen Meng

Round 2

Reviewer 1 Report

Comments and Suggestions for Authors

Accepted in the present form

Reviewer 2 Report

Comments and Suggestions for Authors

.

Comments on the Quality of English Language

.